# Comparing Fisher Information Regularization with Distillation for DNN Quantization

**Prad Kadambi**
Arizona State University
padambi@asu.edu

**Karthikeyan Natesan Ramamurthy**
IBM Research
knatesa@us.ibm.com

**Visar Berisha**
Arizona State University
visar@asu.edu

## Abstract

A large body of work addresses deep neural network (DNN) quantization and pruning to mitigate the high computational burden of deploying DNNs. We analyze two prominent classes of methods; the first class uses regularization based on the Fisher Information Matrix (FIM) of parameters, whereas the other uses a student-teacher paradigm, referred to as Knowledge Distillation (KD). The Fisher criterion can be interpreted as regularizing the network by penalizing the approximate $KL$-divergence (KLD) between the output of the original model and that of the quantized model. The KD approach bypasses the need to estimate the FIM and directly minimizes the KLD between the two models. We place these two approaches in a unified setting, and study their generalization characteristics using their loss landscapes. Using CIFAR-10 and CIFAR-100 datasets, we show that for higher temperatures, distillation produces wider minima in loss landscapes and yields higher accuracy than the Fisher criterion.

## 1  Introduction

Deep Neural Network (DNN) networks are compressed or quantized to reduce hardware complexity, memory usage, and energy usage. A DNN model, $\mathcal{F}(x; \theta)$, has output distribution $p(y|x; \theta)$. Upon quantization, the output of the network is perturbed to $p(y|x; \theta - \delta_\theta)$, where $\theta - \delta_\theta \equiv Q(\theta)$, and $Q(.)$ is the quantization operation. Quantization has a deleterious effect on the performance of the model, particularly in the low-bit regime, where the number of bits $N \leq 4$ [1, 2, 3, 4, 5].

The performance degradation can be mitigated using a regularizer that minimizes the $KL$-divergence (KLD) between a high performance FP32 (32 bit floating point) network, and a quantized network. We consider the FP32 DNN, $\mathcal{F}(x; \phi)$, with output distribution $p(y|x; \phi)$, and a quantized network, $\mathcal{F}(x; Q(\theta))$ with output distribution $p(y|x; Q(\theta))$. We use $p_\phi$ and $p_{Q(\theta)}$ to denote the output distributions. The regularized loss for training the quantized network is

$$\tilde{L}(\theta) = L_{CE}(\theta) + \lambda D_{KL}(p_\phi || p_{Q(\theta)}), \tag{1}$$

where $L_{CE}$ is the standard cross entropy loss. Note that we consider two *different* sets of parameters $\phi$ and $\theta$ here for generality. Broadly speaking, there are two paradigms for operationalizing the regularizer in (1): (a) Setting $\phi = \theta$ and using a second order expansion around $\theta$ of the $KL$-divergence in terms of the Fisher information matrix (FIM), then empirically estimating the FIM and solving for $\theta$; (b) Using distillation by pre-training and fixing an FP32 teacher model with parameters $\phi$; then optimizing the student quantized model with parameters $\theta$.

In this paper, we place these two approaches in a unified setting and study their generalization characteristics using their loss landscapes. We show an advantage for implementing (1) via distillation as increasing the temperature hyperparameter in distillation flattens the loss function at the minimum.

34th Conference on Neural Information Processing Systems (NeurIPS 2020), Vancouver, Canada.

We empirically validate that the flatter minima result in improved model accuracy on benchmark data for different bit widths.

## 2 Methods

In this section we provide an overview of the two families of methods: Fisher-based regularization and KD. The details of regularizing using Fisher Information for DNN quantization is outlined in Section 2.1; Quantization based on KD is covered in Section 2.2. In Section 2.3 we derive bound on the trace of the Hessian, inversely relating KD temperature and loss surface curvature.

### 2.1 Fisher Information regularization for quantization

We return to the KL divergence in (1) and set $\phi = \theta$ (where $\theta$ are the FP32 parameters). This loss function identifies a broad optimum for the network such that small perturbations $\delta_\theta = \theta - Q(\theta)$ around the optimum do not change the output distribution of the network. For sufficiently small $\delta_\theta$, a second order Taylor expansion of the KL divergence can be written in terms of the FIM: $D_{KL}(p_\theta||p_{\theta-\delta_\theta}) \approx \delta_\theta^T \mathbf{F} \delta_\theta$. Therefore, we implement (1) using the FIM,

$$\tilde{L}(\theta) = L_{CE}(\theta) + \frac{\lambda}{2}\delta_\theta^T \mathbf{F} \delta_\theta. \tag{2}$$

The FIM, $\mathbf{F}$, defines a metric on a Riemannian manifold of probability densities on the space of network parameters. The FIM is the expected value of the negative Hessian of the log-loss (standard cross entropy for DNNs), or equivalently, the covariance of the *score*[1]. Many approaches for reducing the complexity of DNNs can be interpreted as means of implementing the regularizer in (2). For network pruning, unnecessary weights can be eliminated using salience criteria based on the FIM (estimated using Hessian) [6, 7, 8, 9, 10, 11]. These methods are heuristics for discarding weights that induce a large $\delta_\theta$ for parameters that have a small entry in $\mathbf{F}$.

Several methods quantize weights based on the FIM. In [12] weights are clustered by FIM diagonal value, and high FIM clusters are assigned larger bit depth. In [13] a non-uniform quantizer is learned using a FIM-weighted $k$-means scheme. Matrix-free power iteration is used for mixed-precision layer-wise quantization based on the Hessian eigenspectrum [14, 15, 16]. These methods shape the quantization error, $\delta_\theta$, such that it is inversely proportional to the entries in $\mathbf{F}$. Some methods directly regularize $\delta_\theta^T \delta_\theta$[17] (equivalent to using an identity matrix for FIM), or constrain $\ell_\infty$-norm of $\delta_\theta$ [5].

**Implementing FIM-based methods:** Approaches that optimize a variant of (2) require estimation of the FIM. We estimate $\mathbf{F}$ using the diagonal approximation in [4, 13, 18, 12]. The second moment, $\hat{v}(\theta)$, of the ADAM optimizer [19] is an exponential moving average of squared gradients, and can be used to estimate the FIM diagonal. Then, the gradient update from the regularizer for weight $\theta_j$ is:

$$\frac{\partial}{\partial \theta_j} \frac{\lambda}{2} \delta_\theta^T \mathbf{F} \delta_\theta = \frac{\partial}{\partial \theta_j} \frac{\lambda}{2} \sum_i F_{ii} \delta_{\theta_i}^2 = \lambda F_{jj} \delta_{\theta_j}. \tag{3}$$

Before regularization, $\mathbf{F}$ is fixed after first finding a good minimum. The regularizer gradient is not used when computing momentum, but only added when calculating the final parameter update [20].

### 2.2 Distillation as $KL$-Divergence minimization

We return to the original loss function based on the KL divergence in (1). In contrast to FIM regularization for quantization, KD uses a pre-trained FP32 teacher, $\mathcal{F}(x; \phi)$, to guide the training of the quantized, reduced capacity student model, $\mathcal{F}(x; Q(\theta))$. We expand (1), $D_{KL}(p_\phi||p_{Q(\theta)}) = H(p_\phi, p_{Q(\theta)}) - H(p_\phi)$; since the teacher model is pre-trained and fixed, the second term $H(p_\phi)$ is a constant. Thus, (1) reduces to the traditional distillation loss,

$$\tilde{L}(\theta) = L_{CE}(\theta) + \lambda H(p_\phi, p_{Q(\theta)}). \tag{4}$$

KD has been previously used for DNN quantization; because KD is parameter agnostic, it can be used with other compression techniques [10, 21, 22, 23, 24, 25]. Quantization-aware KD [26] distills a compressed student model onto an FP32 teacher trained from scratch, before fine tuning the student using the teacher. FIM-based approaches have even been used in conjunction with KD [10] for model pruning. KD has been used extensively to reduce the size of transformer models [22, 23, 24, 25, 27]. These variants of KD compress BERT [28] to 7 MB applying layer-wise knowledge transfer [22], or apply KD every $L$-layers [29].

### 2.3 Flatness of minima

The flatness of the loss function at a minimum is encoded in the eigenvalues of the Hessian and is used by [30, 31] to connect improvements in generalization performance to flatter minima. The

---

[1]*Score* here refers to the partial derivative of the log likelihood with respect to $\theta$.

study in [30] characterizes loss curvature at a minimum using a diagonal FIM to estimate the Hessian eigenspectrum. The FIM-based methods directly make use of the Hessian during optimization; however KD bypasses the need to compute the Hessian altogether. As we detail in the ensuing section, KD consistently outperforms FIM-based methods on multiple benchmark problems and for all bit widths considered. We hypothesize that the increased test accuracy of KD corresponds to reduced curvature of the loss at the minimum. We show that curvature is inversely proportional to KD temperature by establishing an upper bound on the the trace of the Hessian. The Hessian's trace has been applied as a useful measure of curvature in previous studies [15].

To establish the connection between curvature and the temperature parameter, consider the last layer (softmax) of a DNN. KD uses a temperature softmax: $q_k = \frac{e^{z_k/T}}{\sum_j e^{(z_j/T)}}$ to smooth predicted class probabilities as a function of $T$ (output probability $q_k$ for the $k^{th}$ class). For a $K$ class problem, the softmax layer has weights $\theta^\ell \in \mathbb{R}^{K \times M}$, layer inputs $x^\ell \in R^M$, and the pre-softmax logit for the $k$-th class is $z_k = \theta^{\ell^T}_{k,:} x^\ell$, where $\theta^\ell_{k,:}$ is a row of the weight matrix. KD temperature plays an important role for quantization as it directly impacts the width at the optimum and, as a result, the allowable quantization error. The Hessian $\mathbf{H} \in \mathbb{R}^{MK \times MK}$ has block structure with each $M \times M$ block,

$$\mathbf{H}_{kj} = \frac{\partial^2 L_{CE}(\theta)}{\partial \theta^\ell_{k,:} \partial \theta^\ell_{j,:}} = \sum_{n=1}^{N} \frac{1}{T^2} q_{nj}(I_{kj} - q_{nk}) x^\ell_n x^{\ell^T}_n, \tag{5}$$

where $I_{k,j}$ is an element of the identity matrix, $k, j \in \{1, \ldots, K\}$, and additional the subscript $n$ denotes the sample index out of $N$ data points. The trace of the Hessian can be expressed as

$$\text{Tr}(\mathbf{H}) = \sum_{k=1}^{K} \sum_{n=1}^{N} \frac{q_{nk}(1 - q_{nk})}{T^2} \text{Tr}(x^\ell_n x^{\ell^T}_n) \le \frac{M(K-1)}{T^2 K} \sum_{n=1}^{N} \text{Tr}(x^\ell_n x^{\ell^T}_n), \tag{6}$$

where the bound the arises from $q_k(1 - q_k) \le \frac{K-1}{K^2}$. This bound shows an inverse relationship between the curvature of the loss at the minimum and the temperature parameter, $T^2$. That is, as the temperature is increased, the minimum becomes flatter. This is validated experimentally in the next section where we visualize the loss at the minimum. Furthermore, we also show that the flatter minimum results in improved accuracy.

## 3 Results and Discussion

In this section we empirically compare methods based on FIM regularization with distillation on CIFAR-10 and CIFAR-100. We first visualize the loss landscapes for a 4-bit quantized ResNet-18 on CIFAR-10, then compare accuracy results across different bit widths on CIFAR-10 and CIFAR-100. For all bit width levels on CIFAR-10 and CIFAR-100, networks were trained for 300 epochs with a straight through estimator (STE) to find a good minimum $\theta^*$, and then subsequently trained with the regularizers (and STE) for 100 epochs. We also include mean squared quantization error (MSQE) regularization as a useful comparison as it is identical to Fisher regularization, as it assumes that the FIM is the identity matrix (i.e. assuming all parameters are equally important).

### 3.1 Loss landscape visualization

The filter normalization method of [32] is used to compare the loss surfaces at the minimum for each regularization method. The loss is plotted along two random vectors $\beta$ and $\gamma$, sampled from a Gaussian distribution and rescaled per filter. Qualitatively, Figure 1 shows that the loss surface exhibits local convexity along $\beta$ and $\gamma$, and the improvement in test accuracy correlates with minimum width. This is also seen in [32] where increases in test accuracy (due to modifications such as L2 regularization, skip connections, selection of hyperparameters), correspond to flatter minima. To quantitatively characterize minima flatness, we fit a paraboloid to the computed loss surface as it is locally convex. The trace of the Hessian of the *fitted paraboloid* is used to measure the curvature of the loss. For consistency, $\beta$ and $\gamma$ are fixed across all regularization methods, rescaled with filter normalization for every network. Figure 1 (h), demonstrates that as temperature is increased, test accuracy increases, and the trace of the Hessian of the fitted curve decreases. This empirically consistent with the bound in (6). This is repeated for 10 randomly chosen $\gamma$ and $\beta$.

### 3.2 Accuracy results

Tables 1 (a) and (b) show the effect of regularization on CIFAR-10 and CIFAR-100 across all bit widths. MSQE and Fisher regularization not only result in similar test accuracy, but have very similar minima flatness. Since networks trained with STE already use $Q(\theta^*)$ in the forward pass, reducing the perturbation $\delta_{\theta^*}$ does not change the operating point of the network (for both Fisher and MSQE).

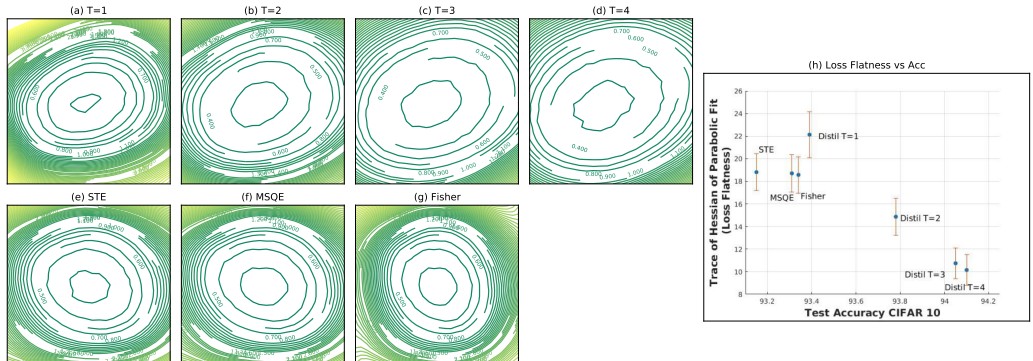

Figure 1: (a)-(g) Test loss surface on CIFAR-10 of 4-bit quantized ResNet-18. Around the minimum, the loss surface is locally convex about the two random directions $\gamma$ and $\beta$ that are used to visualize the loss. (h) shows CIFAR10 test accuracy vs flatness of minimum as measured by the trace of the Hessian of a parabolic curve fit to the local loss landscape.

| Bits A/W | STE | MSQE | Fisher | KD |
|---|---|---|---|---|
| 32/2 | 93.5±.2 | 93.6±.2 | 93.6±.2 | **94.0±.3** |
| 4/4 | 93.4±.2 | 93.4±.2 | 93.4±.2 | **94.2±.1** |
| 4/1 | 90.7±.4 | 91.3±.3 | 91.2±.2 | **91.4±.2** |
| 2/2 | 89.3±.2 | 89.4±.2 | 89.4±.4 | **89.9±.2** |

(a)

| Bits A/W | STE | MSQE | Fisher | KD |
|---|---|---|---|---|
| 4/4 | 72.0±.5 | 72.3±.2 | 72.1±.4 | **73.6±.2** |
| 4/2 | 67.9±.3 | 69.7±.5 | 70.0±.3 | **70.4±.3** |

(b)

Table 1: Performance of quantized models (A: Activation, W: Weights). (a) Test Accuracy of ResNet-18 on CIFAR10 (average over 5 trials). Distillation temperature of T=4 is used. Mean squared quantization error (MSQE) corresponds to setting an identity matrix for the FIM. STE denotes training with straight-through estimator and serves as a baseline. FP32 baseline accuracy (teacher) is 94.4%. (b) Quantized ResNet-18 accuracy for Cifar100. FP32 Baseline (teacher) is 74.1%.

### 3.2.1 Why does distillation outperform Fisher-based methods?

We posit that the difference in performance between distillation, Fisher, and MSQE regularization is a consequence of the difference in minima flatness observed in Figure 1 (h). The temperature, $T$, in distillation widens the minimum, as measured by the trace of the Hessian, which translates to improved accuracy at all bit widths. Another possible reason for the improved performance is that distillation bypasses FIM estimation altogether, while still minimizing (1). Estimating the FIM is challenging. Approaches often use diagonal approximations [14, 15, 18, 12, 13] (the Hessians of DNN loss surfaces are non-diagonal [33, 34]), or computationally intensive methods [11, 14, 15]. These approaches have tenuous theoretical foundation, since the "empirical Fisher approximation," the widely used FIM estimate, computes expectation over the training labels $y_n$, and is distinct from estimating the true FIM, an expectation over model outputs, $y$. Weight updates preconditioned with empirical Fisher can be orthogonal or opposite to updates preconditioned with the true Fisher [35].

## 4  Conclusion

In this work we studied two regularization methods for DNN quantization that constrain the $KL$-divergence between the output distribution of the full precision model and the output of the model under a quantization perturbation. The first method uses an approximation to the $KL$-divergence in terms of the FIM to constrain the solution space. The second method uses distillation as an alternative to directly regularizing the $KL$-divergence between an FP32 and quantized model without assumptions of closeness in parameter space or having to estimate the FIM. Experiments on CIFAR10 and CIFAR100 demonstrated distillation outperformed Fisher and MSQE regularization. We provided evidence that this improvement is due to flatter minima obtained from increasing the temperature parameter in KD.

## Broader Impact

An extremely broad array of applications demand the use of low power DNNs across computer vision, natural language processing, and speech recognition. Computer vision systems such as autonomous driving, human activity detection, robotic systems require efficient DNN inference. Voice driven smart home systems also require low latency, low power deep learning inference for understanding spoken commands.

Beyond mobile devices, efficient and accurate DNN inference is likely to be crucial for datacenters as well. Datacenter workloads can involve DNN inference for recommender systems, computer vision, speech processing, and language. An increasing and large fraction of datacenter workload is expected to come from DNN inference [36] while the model size of state of the art models, in key application domains such as natural language processing, are simultaneously also increasing [28, 37, 38, 39]. DNN compression could become a necessity for low-latency inference of these complex models, and for managing energy density and ecological impact.

Several ethical concerns arise in the application areas of low power DNN. Pervasive sensor networks could utilize efficient, high performance DNN models for to track individual behaviors, movements, and intents without the knowledge or consent of the surveilled. For example, The authors in [10] use the Fisher information and Knowledge distillation to prune a network for faster gaze prediction.

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
