# OpenReview forum: "Comparing Fisher Information Regularization with Distillation for DNN Quantization"
_NeurIPS.cc/2020/Workshop/DL-IG — NeurIPSW 2020: DL-IG Poster_

### Official Review · AnonReviewer2 · 2020-10-25
**Review of "Comparing Fisher Information Regularization with Distillation for DNN Quantization"**

**Rating:** 7
**Confidence:** 4

**Review:**

This paper compares the loss landscape for two different schemes to quantizing neural networks. Although I wasn't very familiar with work in this area, I found the introduction very clear and appreciated the thoroughness of the related work.

Eq. 6 seems intuitive, in the high temperature limit we get a uniform prediction so there is no additional loss from discretization. Does the bound actually get used? It isn't necessary for the qualitative remarks and doesn't seemed to be used in the results, unless I missed something. (Maybe it is helpful because otherwise T implicitly appears in some terms in the sum.)

The preliminary result is interesting, and seems to show the advantage of high temperature distillation. The emphasis of most quantization schemes is to maintain a reasonable loss (cross entropy in this case) while discretizing the parameters. You didn't show the cross entropy, but my guess is that it fares much more poorly than the test accuracy. By focusing on the accuracy, you are sort of introducing some prior knowledge into the problem, that in practice we care more about prediction accuracy than cross entropy. By increasing the temperature, it makes it easier to find a good discretization that will probably give the same prediction, but with less confidence, and this will disproportionately affect the cross entropy loss, but not the accuracy. I don't see this as a problem, I just would like to see more discussion of this in a longer version of this interesting work.  (Just to emphasize, I'm claiming that the key feature here is the softmax temperature - which happens to be a good way to de-emphasize confidence without changing accuracy AND which happens to lead to flatter minima. I wasn't convinced that flatter minima themselves would necessarily lead to better accuracy.)

---

### Official Review · AnonReviewer1 · 2020-11-01

**Rating:** 7
**Confidence:** 4

**Review:**

This paper discusses two quantization methods, one where the quantized model is initialized at the true model and trained further with a penalty that depends upon the Fisher Information Matrix (FIM), and another where a teacher is distilled into a quantized student using knowledge distillation (KD) techniques. While the two objectives are close to each other, the FIM is the Hessian of the KL-divergence of the conditional probability distribution, the paper shows that KD achieves a marginally better accuracy on CIFAR-10 and CIFAR-100.

How do you initialize KD in eq. 4? KD-based quantization cannot be compared with FIM-based quantization because the former involves training a student from scratch while the latter initializes the quantized model at the teacher’s weights. Further, it would be interesting to quantify the distance between the original model and the quantized model, KD (if initialized at the true model) allows more freedom to the quantized parameters than the FIM-based method.

---

### Author Response · Authors · 2020-12-12
**Additional Contant**

A 5 minute video presentation of this work is available here: https://www.youtube.com/watch?v=bDFL-VSlul8&feature=youtu.be

---

### Decision · Program_Chairs · 2020-11-07

Accept (Poster)